# Effects of Frying Conditions on Volatile Composition and Odor Characteristics of Fried Pepper (*Zanthoxylum bungeanum* Maxim.) Oil

**DOI:** 10.3390/foods11111661

**Published:** 2022-06-06

**Authors:** Ruijia Liu, Nan Qi, Jie Sun, Haitao Chen, Ning Zhang, Baoguo Sun

**Affiliations:** Beijing Key Laboratory of Flavor Chemistry, Beijing Technology and Business University, Beijing 100048, China; lrj990806@163.com (R.L.); sqinan@126.com (N.Q.); chenht@th.btbu.edu.cn (H.C.); zh_ningts@btbu.edu.cn (N.Z.); sunbg@btbu.edu.cn (B.S.)

**Keywords:** fried pepper oil, frying temperatures, frying times, GC–MS, odor, aroma profiles

## Abstract

Fried pepper (*Zanthoxylum bungeanum* Maxim.) oil (FPO) is widely used in Chinese cuisine because of its unique aroma. To investigate the effects of different frying temperatures and different frying times on the volatile composition and odor characteristics of FPOs, descriptive sensory analysis (DSA), solvent-assisted flavor evaporation–gas chromatography–mass spectrometry (SAFE–GC–MS) and electronic nose (E-nose) were used to analyze the FPOs (FPO1–FPO4 represented the pepper oil fried at 110 °C, 120 °C, 130 °C, and 140 °C; FPO5–FPO7 represented the pepper oil fried for 10 min, 20 min and 30 min). The results showed that FPO3 and FPO6 had strong citrus-like and floral aromas and exhibited significant advantages in sensory attributes. A total of 46 volatile compounds were identified by SAFE–GC–MS; among them, FPO3 and FPO6 had a higher volatile compound content. *β*-Caryophyllene was detected in only FPO3 and FPO6; linalool was higher in FPO3 and FPO6, which might cause them to exhibit stronger floral and citrus-like aromas. The presence of (2*E*,4*E*)-2,4-decanedienal would be one of the reasons for the strong fatty aroma exhibited in FPO4 and FPO7. FPO3 and FPO6 were associated with citrus-like and floral aromas by partial least squares regression (PLSR) analysis, which agreed with the sensory evaluation results.

## 1. Introduction

*Zanthoxylum bungeanum* Maxim., a variety of a genus within Rutaceae, consists of approximately 250 species of trees and shrubs from all over the world, including 45 species and 13 varies in China [1,2]. In China, *Zanthoxylum bungeanum* Maxim. is also known as Chinese pepper, or “huajiao”. Huajiao not only has a unique aroma and taste, but it also has a high medicinal value [3]. Therefore, Chinese people add huajiao as an important spice in Chinese cuisine, as well as treating it as an herbal medicine. The history of pepper cultivation in China can be traced back to about 3000 years ago. From the northwest to the southwest of China, there are many famous pepper-growing areas, such as Hancheng in Shaanxi and Hanyuan in Sichuan, whose cultivars include red pepper and green pepper [4]. The aroma quality of Chinese pepper is considered to be influenced by the soil, light conditions, and temperature in the growing places.

Frying is the most common way to prepare Chinese pepper oil in Chinese dishes. The special aroma of fried food is the result of a complex interaction between food compounds and the during the heating process [5,6]. The frying process produces volatile compounds through chemical reactions (e.g., lipid oxidation, polymerization, and hydrolysis), resulting in a special aroma [7]. It has been shown that frying time and frying temperature are important parameters affecting the odor of fried products [8]. Therefore, these two factors can be controlled to prepare a good aroma of fried pepper oil (FPO).

There is no doubt that the attractive aroma is an important reason for its popularity. Stoichiometry is an effective method for screening processing conditions. Previous research has used gas chromatography–mass spectrometry (GC–MS) and descriptive sensory analysis (DSA) combined with partial least squares regression (PLSR) to analyze the volatile compounds and aroma attributes of almonds roasted at two different temperatures, leading to the identification of substances that can predict consumer preferences [9]. In addition, a research team also used GC–MS combined with principal component analysis (PCA) and orthogonal partial least squares discriminant analysis (OPLS–DA) to identify and compare the volatile compounds in shallot oil prepared at five different frying temperatures. The best quality of shallot oil was obtained by frying at 150 °C. Multivariate data analysis is important for the selection of the optimal temperature for fried shallot oil [10]. Besides, GC–MS, DSA, and E-nose have been used to find differences in the aroma of FPO at different temperatures, and 100 °C is the optimum frying temperature. They have discovered the key substances affecting its aroma through aroma recombination and omission experiments [11].

For the study of food aromas, it is crucial to choose the appropriate extraction method [12]. Volatile compounds can be quickly and efficiently separated from solvent extracts of foods, oil samples, and even fruit pulps using solvent-assisted flavor evaporation (SAFE) [13]. Currently, SAFE is widely used in research on the aromas of wine, dairy products, and meat products [14,15]. E-nose is a device that combines sensor technology, electronics, biochemistry, and artificial intelligence. It simulates the human olfactory system to measure and characterize volatile compounds [16,17,18,19]. Although there have been previous studies on the volatile compounds in FPO, there are fewer studies that evaluate the effects of the frying process on the aroma of FPO.

Thus, the objective of this study was to evaluate effects of the frying process on the aroma and sensory profiles of FPO by DSA, E-nose, and SAFE–GC–MS analysis. The results will provide a guide for selecting appropriate frying processes to produce high-quality pepper oil. This study can improve the quality of pepper oil products and provide guidance for industrial FPO products.

## 2. Materials and Methods

### 2.1. Chemical Standards

Dichloromethane (≥99%), sodium sulfate, 2-hydroxy-3-(octanoyloxy)propyl decanoate (≥98%), (*E*,*E*)-2,4-hexadienal (>95%), (*E*,*E*)-2,4-heptadienal (>98%), and hydroxyacetone (90%) were purchased from Mreda (Beijing, China). 1,2-Dichlorobenzene (99%), sabinene (≥95%), *δ*-3-carene (≥90%), p-cymene (99%), *β*-ocimene (95%), terpinolene (95%), octanal (98%), carveol (95%), piperitone (95%), *α*-terpineol (≥95%), geranyl acetate (≥95%), (*E*)-2-decenal (95%) linalyl acetate (98%), (*E*)-limonene oxide (97%), dihydrocarvone (98%), neryl acetate (98%), *β*-terpineol (≥95%), (-)-carveol (≥95%), and n-alkanes (C6~C26) were purchased from Sigma-Aldrich (Shanghai, China). *β*-Caryophyllene (≥95%) and nonanoic acid (98%) were purchased from Aladdin (Shanghai, China). (2*E*,4*E*)-2,4-decanedienal (95%) was purchased from Acros (Beijing, China). Terpinyl acetate (≥97%), 2-acetyl pyrrole (99%), decanoic acid (99%), octanoic acid (≥99.5%), and 2-acetylfuran (99%) were purchased from Macklin (Beijing, China). Copaene (96%), linalool oxide (97%), and perillen (95%) were purchased from Trc (Shanghai, China). 1-Octen-3-ol (98%), phenylethyl alcohol (99%), and 5-methyl furfural (98%) were purchased from Beijing Sinopharm Chemical Reagent Co., Ltd. (Beijing, China). Hotrienol (≥98%) and perillalcohol (≥95%) were purchased from Shanghai Acmec Biochemical Co., Ltd. (Shanghai, China). 1,8-Cineole (≥98%), limonene (≥95%), linalool (≥95%), *β*-myrcene (≥95%), and (*E*)-2-heptenal (≥95%) were purchased from J & K Scientific Co., Ltd. (Shanghai, China).

### 2.2. Materials

Samples were taken from Maoxian red peppers grown in Maoxian County, Sichuan Province, China. Fresh red peppers were dried by hot air drying at 40–60 °C for 10 h. The corn germ oil used to fry the pepper oil was bought from Jin Long Yu Co., Ltd. (Shanghai, China). Dried red peppers and corn germ oil were used in this investigation.

### 2.3. Sample Preparation of FPOs with Different Frying Temperatures

Corn germ oil (150 g) was added to a 0.5 L round-bottom flask and heated in an oil bath with various setting temperatures (110 °C, 120 °C, 130 °C, and 140 °C). The weighed pepper granules (50 g) were put into a round-bottom flask and fried at a constant temperature for 20 min, being stirred continuously during the process to prevent local overheating. After 20 min, the pepper granules were fished out, and the FPOs were cooled to room temperature in an ice-water bath and then sealed and stored under refrigeration for subsequent analysis. The FPOs prepared at 110 °C to 140 °C were named FPO1 to FPO4. Three parallel tests were conducted for each sample.

### 2.4. Sample Preparation of FPOs with Different Frying Times

Corn germ oil (150 g) was added into a 0.5 L round-bottom flask, and pepper granules (50 g) were added once the corn germ oil had been heated to 130 °C. The pepper granules were fried at a constant temperature for 10 min, 20 min, and 30 min, with constant stirring during the heating process to prevent local overheating, and the pepper granules were fished out at the end of heating. The FPOs were cooled to room temperature in an ice-water bath and then sealed and stored under refrigeration for subsequent analysis. The FPOs prepared at 10 min to 30 min were named FPO5 to FPO7. Three parallel tests were conducted for each sample.

### 2.5. Sensory Analysis

The sensory evaluation was conducted by 15 team members (7 men and 8 women) from the Key Laboratory of Flavor Chemistry, Beijing, China. The sensory evaluation team members were trained in quantitative descriptive analysis (QDA). Assessment was carried out in the sensory laboratory at a temperature of 25 °C. The samples were prepared 1 h in advance and placed in a water bath at 25 °C. FPO (5 g) was poured into a transparent, odorless PET bottle with a lid (volume = 50 mL). In the training process, the team members first gave sensory descriptors to describe the aroma of a FPO, and then recorded statistics on the given descriptors to select the most frequent descriptors as quantitative descriptors. Finally, five descriptors were obtained, namely, herbal, citrus, floral, rosin, and fatty aroma. These descriptors were compared with the following reference odorants for 2-hydroxy-3-(octanoyloxy) propyl decanoate solutions: 1,8-cineole (herbal, 1%), limonene (citrus-like, 1%), linalool (floral, 1%), *β*-myrcene (rosin, 1%), and (*E*)-2-heptenal (fatty, 1%). Then, the sensory evaluators scored the FPOs for odor only. The intensities were performed by linear scoring with a scale from 0 (not detected) to 5 (very intense odor), with increments of 1.0.

### 2.6. E-Nose Analysis

The Portable Electronic Nose System (PEN 3, Airsense, Berlin, Germany) contains 10 metal oxide semiconductors for aroma information and volatile compound data. Each sensor has a certain sensitivity to each characteristic volatile compound. Each sample (0.1 g) was put into a 30 mL bottle. The cap was sealed with plastic wrap and balanced in a 25 °C water bath for 10 min to reduce sensor drift caused by environmental changes. Gas was then injected into the detection system through the probe from the headspace of the sample at a flow rate of 300 mL/min. The response of the sensor was measured by the G/G_0_ ratio. (G and G_0_ represent the conductance of the sensor to sample gas and clean gas, respectively.) The sampling time was 150 s, which made the sample data gradually stable. Three parallel tests were conducted for each sample.

### 2.7. Extraction of the Volatile Compounds by SAFE

FPO (50 g) was dissolved in dichloromethane (200 mL) and sealed by shaking at 120 rpm (grant OLS200, Cambridgeshire, UK) for 0.5 h. In addition, as the internal standard (8.2 mg/L, 20 μL), 4-octanal was added before the extraction produce. The solutions were subjected to high-vacuum distillation using SAFE. The extracted solution was dried with anhydrous Na_2_SO_4_ and filtered. Then, the distillate was concentrated to 1 mL using a Vigreux column. After sealing, the final distillate was stored in a refrigerator at −20 °C for GC analysis.

### 2.8. GC–MS Analysis

GC–MS analysis was performed by a Thermo Fisher Trace 1310 gas chromatograph (Thermo Fisher Scientific, Waltham, MA, USA) coupled with a Thermo Fisher mass spectrometer (Thermo Fisher Scientific, Waltham, MA, USA). A TG-WAX column (30 m × 0.25 mm i.d., 0.25 μm, Thermo Fisher Scientific, Waltham, MA, USA) was used for separation, and helium was used as the carrier gas. The separation was carried to the chromatographic column at a fixed flow rate of 1.0 mL/min. The initial temperature was 40 °C for 1 min, which was then increased to 140 °C at a rate of 2 °C/min, followed by a 1 min hold. Finally, it was increased to 220 °C at a rate of 6 °C/min and held for 1 min. Mass detector conditions were as follows: ionization energy, 70 eV; transmission line temperature, 250 °C; ion source temperature, 250 °C; mass range, *m*/*z* 45–350; and solvent delay, 5 min.

### 2.9. Identification and Quantitation of Aroma Compounds

Volatile compounds were identified by comparing their retention indices (RIs), mass spectra, and standard reference compounds. A retention index (RI) was calculated for each volatile compound using the retention times of a homologous series of C_6_–C_30_ n-alkanes. Mass spectral comparisons were based on the NIST 14 mass spectral database. Volatile compounds in each FPO were semiquantitated on the basis of 4-onctanal as an internal standard. The concentrations of the aroma compounds were calculated from the ratio of the peak areas and the concentration of 4-octanal. In order to verify the accuracy of the quantitation method, the recovery of 4-octanal was calculated. The recovery calculated was 97.2%.

### 2.10. Statistical Analysis

The experimental results were expressed as the mean ± standard deviation (SD) and analyzed by one-way analysis of variance (ANOVA) using SPSS-IBM 19.0 software. Tukey’s post hoc test (*p* < 0.05) was performed to compare means and samples that were significantly different. A spider diagram for sensory attribute scores was drawn using Origin 2018 software (OriginLab, Northampton, MA, USA). Principal component analysis and partial least squares analysis were performed using XLSTAT v. 2018 (Addinsoft, New York, NY, USA).

## 3. Results and Discussion

### 3.1. Effect of Frying Temperature on Aroma and Sensory Profiles of FPO

Analysis of the overall aroma profile differences among the FPOs with different frying temperatures was performed by DSA. The significant differences were analyzed as shown in Appendix A. As seen in Figure 1, there were significant overall aroma differences among the 4 FPOs. The intensity of the fatty and herbal aromas gradually increased significantly with frying temperature and exhibited the highest levels in FPO4. This might be attributed to the increase in the production of compounds with a fatty aroma due to the increase in frying temperature. Among the other FPOs, FPO1 exhibited a prominent rosin aroma. In addition to this, FPO3 exhibited the most intense citrus-like and floral aromas, which made its overall aroma more harmonious and acceptable to the sensory evaluation team.

The E-nose has a wide range of applications in aroma identification, which can express the finishing aroma profile presented by volatile compounds to the samples, rather than the qualitative and quantitative results. In order to better display the aroma differences of FPOs at different frying temperatures, PCA was used to perform mathematical statistics on the results of the E-nose analysis. As shown in Figure 2, the first principal component (F1) comprised 71.46% of the information of the total variable, and the second principal component (F2) comprised 19.34%. The total contribution rate of the two principal components was 90.80%, indicating that the analysis results can illustrate the main characteristic information of the sample. The difference in aromas between FPOs was indicated by the distance of the graph; the farther the distance, the greater the difference in their aromas. Obviously, there were significant differences in the aroma profiles of the FPOs with different frying temperatures. Among them, F1 clearly distinguished FPO4 from the other 3 FPOs, and FPO4 appeared on the negative side of F1, while the other 3 FPOs appeared on the positive side of F1, indicating that there were significant differences in the aroma of FPO4 compared to that of the other FPOs. As shown in Figure 1, FPO4 was significantly distinct from the other FPOs in floral, fatty, rosin, herbal, and citrus-like aromas. This indicated that the results of the E-nose principal component analysis were consist with the sensory evaluation. While the natural senses lack objectivity and standardization and are easily affected by factors such as the environment and individual differences, the E-nose can simulate the biological olfactory system but cannot completely replace the natural sense of smell, so the sensory results of the two methods can mirror each other and complement each other.

As the results of the application of SAFE on FPOs prepared at different frying temperatures, the volatile compounds of pepper oil are shown in Table 1. A total of 46 volatiles, distributed among 7 chemical classes, were identified, including 12 terpenoids, 4 ketones, 14 alcohols, 7 aldehydes, 3 carboxylic acids, 4 esters, and 3 heterocyclic compounds. Among them, the alcohols were the most diverse, followed by the terpenoids. In the 4 groups of FPOs, 16 aroma compounds were detected together, which could constitute the basic aroma profile of pepper oil. There were 30 different compounds in the different FPOs, indicating that the frying temperature had a great influence on the overall aromas of the FPOs. Among them, (*E*,*E*)-2,4-hexadienal and *β*-caryophyllene were only detected in FPO3. This might be one of the reasons that led FPO3 to exhibit the strongest floral and citrus-like aromas in the sensory evaluation results of Figure 1. With the increase in frying temperature, the total volatile compound content of the FPOs changed significantly. Compared with the FPOs prepared at lower frying temperatures (FPO1: 214.53 mg/kg and FPO2: 293.07 mg/kg), the total amounts increased in FPO3 (516.67 mg/kg), and then decreased in FPO4 (321.95 mg/kg). This showed that FPO3 had the highest amounts of volatiles in terms of both kind and content. This might be mainly due to the fact that the volatiles of the pepper have not been fully dissolved into the oil at the lower frying temperatures, and therefore, some of the volatiles, especially the terpenes and terpenoids, were low in FPO1 and FPO2. As the frying temperature continued to increase to 140 °C, these volatiles might have transformed into the other compounds through complex interactions with lipid oxidation or decomposition, thus resulting in their content being significantly reduced [20]. For example, limonene and linalool were oxidized to (*E*)-limonene oxide and linalool oxide, respectively [21]. Therefore, the kinds and contents of volatiles were the highest in FPO3. Volatiles, such as (2*E*,4*E*)-2,4-decanedienal, were only detected in FPO4. They are produced by the peroxidation of linoleic and arachidonic acids during the heat treatment of foods [22]. Due to their low odor-detection thresholds (0.027 mg/kg), when a small amount was present, (2*E*,4*E*)-2,4-decanedienal had a good frying aroma. On the contrary, an increase in their concentrations would bring a strong oily taste to the fried pepper oil, which might be one of the reasons for the strongest fatty aroma being exhibited at FPO4 in the sensory evaluation results of Figure 1.

### 3.2. Effect of Frying Times on the Aroma and Sensory Profiles of FPOs

Sensory evaluation analysis was conducted on the overall aroma of fried pepper oil prepared at different frying times (10 min, 20 min, and 30 min); the significant difference analysis is shown in Appendix A, and the results are shown in Figure 3. FPO5 exhibited the weakest fatty aroma as well as the strongest rosin aroma, which might be caused by the short frying time, resulting in the low production of compounds presenting a fatty aroma. In contrast, the aroma profiles of the FPOs changed significantly as the frying time was increased. FPO7 exhibited the strongest herbal and fatty aroma, as well as the weakest rosin aroma, possibly due to the long frying time, which resulted in the high production of compounds presenting herbal and fatty aromas, and the oxidation or degradation of compounds presenting a rosin aroma. On the other hand, FPO6 exhibited floral and citrus-like aromas with a more harmonious overall aroma profile that was more acceptable to the sensory evaluation group.

To investigate the aroma differences between the FPOs prepared at different frying times, the FPOs were subjected to E-nose analysis, and the results are shown in Figure 4. The total contribution rate of the 2 principal components (PC1 and PC2) was 99.94%, which indicated that the analysis results could explain the main characteristic information of the sample. The aroma profiles of different FPOs were far apart from each other and located in different quadrants of the figure, indicating that there were differences in the overall aroma between the FPOs.

The volatile compounds of FPOs prepared with different frying times were analyzed by SAFE–GC–MS. A total of 45 volatile compounds were characterized, including 12 terpenoids, 4 ketones, 13 alcohols, 7 aldehydes, 3 carboxylic acids, 4 esters, and 2 heterocyclics in 3 kinds of FPO with different frying times, as shown in Table 2. In particular, the largest number of alcohol compound species was found, followed by terpenoids. In FPO5, the total content of volatile compounds was 476.96 mg/kg. When the frying time was increased to 20 min, the total content of volatile compounds reached the highest level (FPO6: 516.59 mg/kg). As the frying time continued to be extended, the content of volatile compounds in FPO7 (405.66 mg/kg) showed a decreasing trend. The volatile compounds of the FPOs with different frying times showed a certain difference.

As can be seen in Figure 5, with the increase in frying temperature, the content of various volatile compounds changed more significantly, especially terpenoids, alcohols, and esters. The content of terpenoids showed a gradual decrease with the increase in frying time. Most of the terpenoids, such as *δ*-3-carene, copaene, *β*-myrcene, limonene, and so on, originated from the pepper fruits [23]. When the frying time was short, these compounds were released from the pepper fruits and dissolved in the oil. With longer frying times, these times could either undergo chemical reactions, such as oxidation or degradation, and thus transform into other substances, or they could volatilize into the air, thus causing losses. Some of the alcohols, such as 1,8-cineole and linalool, in the FPOs originated from the pepper fruits [24,25], and other alcohols were derived from the oxidative decomposition of unsaturated fatty acids and the positional isomerization of double bonds on the carbon chain during deep frying [20,26]. For instance, 1-octen-3-ol (which has a mushroom-like odor) was formed from the decomposition of linoleic acid through hydrogen peroxide [27], which had a relatively low human odor threshold of 0.0015 mg/kg, and this had some influence on the overall aroma of the FPO. During the frying process, certain compounds in the pepper fruits could have promoted the oxidation of unsaturated fatty acids in the oil; therefore, the content of the alcohols showed a gradual increase. The esters were derived from the esterification reaction between the alcohols and acids, and the content of the esters increased gradually with the reaction for a certain period of time. Excessive frying time might lead to the decomposition of esters into other compounds, which could result in a lower content. Fat oxidation and the secondary decomposition of its oxidation products could have produced acid, but their contribution to the aroma of FPO was insignificant [28]. Aldehydes, including saturated and unsaturated aldehydes, which were the major volatiles produced by lipid oxidation during frying, showed a tendency to increase in content with time as the reaction occurred.

Alternatively, *β*-myrcene was detected in all 3 FPOs, and the concentration was highest in FPO5 (91.22 ± 1.94 mg/kg), which decreased with time. *β*-Myrcene had a rosin aroma with a very low human odor threshold (0.0012 mg/kg), and this might be one of the reasons for the strongest rosin aroma characteristic being exhibited by FPO5, as shown in Figure 3. By contrast, (2*E*,4*E*)-2,4-decanedienal was detected only in FPO7, probably contributing to the most prominent fatty aroma in Figure 3. In previous studies, 1,8-cineole (herbal aroma) was shown as the key aroma compound that influenced the aroma of FPO [4], with an odor threshold of only 0.0011 mg/kg. The concentration of 1,8-cineole gradually increased with frying time and reached the highest level in FPO7, which probably resulted in the herbal aroma being most prominent, as shown in Figure 3. *β*-Caryophyllene and linalool, with low thresholds and a floral aroma, reached the highest concentrations in FPO6; hence FPO6 presented the strongest floral aroma, as shown in Figure 3. In particular, *β*-caryophyllene was detected only in FPO6, also indicating that frying time had a greater influence on the volatile compounds of the FPOs.

### 3.3. Correlation between Volatile Components and Sensory Attributes of Pepper Oil by Different Frying Processes

In order to more clearly compare the aroma profiles of pepper oils prepared using different frying processes, 46 volatile compounds (Table 1) and sensory attributes were analyzed using PLSR. As shown in Figure 6, both the X variable (relative volatile compound content) and Y variable (intensity of sensory attributes) were located in the circle (r^2^ = 100%; r^2^ represents the degree of interpretation). The proximity of the sensory attributes and the volatile compounds to each other indicated a strong correlation. As shown in Figure 6, the herbal and fatty aromas are located in the upper-right corner of the load diagram, and they exhibit correlations with FPO4 and FPO7, while correlations with octanal, terpinolene, (*E*)-limonene oxide, 2-acetylfuran, and (*E*,*E*)-2,4-heptadienal are evident. The rosin aroma is located in the lower-right region of the load diagram and is the main aroma characteristic of FPO1, FPO2, and FPO5, correlating significantly with compounds such as *β*-myrcene, sabinene, and (+)-Isomenthol. The floral and citrus-like aromas are located in the lower-left region of the load diagram and correlate significantly with compounds such as linalool and linalool oxide. FPO1 and FPO2 are located in the same region, and they exhibited a strong rosin aroma. Additionally, FPO3 and FPO6 were associated with citrus-like and floral aromas, which was consistent with the E-nose analysis results.

## 4. Conclusions

In this paper, the effects of different frying processes on the sensory attributes and volatile compounds of FPOs were investigated by DSA, E-nose, and SAFE–GC–MS. The results showed that FPO3 (with a frying temperature of 130 °C) and FPO6 (with a frying time of 20 min) had the most prominent citrus-like and floral aromas, as well as more harmonious overall aroma profiles. A total of 46 volatile compounds were identified, among which alcohols and terpenes were the most diverse. The total contents of the volatile compounds all showed a trend of first increasing and then decreasing with the increase in frying temperature and time, and these reached maximum values in FPO3 and FPO6, with 516.67 mg/kg and 515.59 mg/kg, respectively. Among them, *β*-caryophyllene was detected only in FPO3 and FPO6; linalool was higher in FPO3 and FPO6, which might be one of the reasons for the strong floral aroma, as compared to the other FPOs. Apart from that, (*E*,*E*)-2,4-hexadienal was detected in FPO3 and was highest in FPO6. The presence of (2*E*,4*E*)-2,4-decanedienal caused FPO4 and FPO7 to exhibit the strongest fatty aroma. The results of PLSR showed that FPO3 and FPO6 were associated with citrus-like and floral aromas, which agreed with the sensory evaluation results.

## Figures and Tables

**Figure 1 foods-11-01661-f001:**
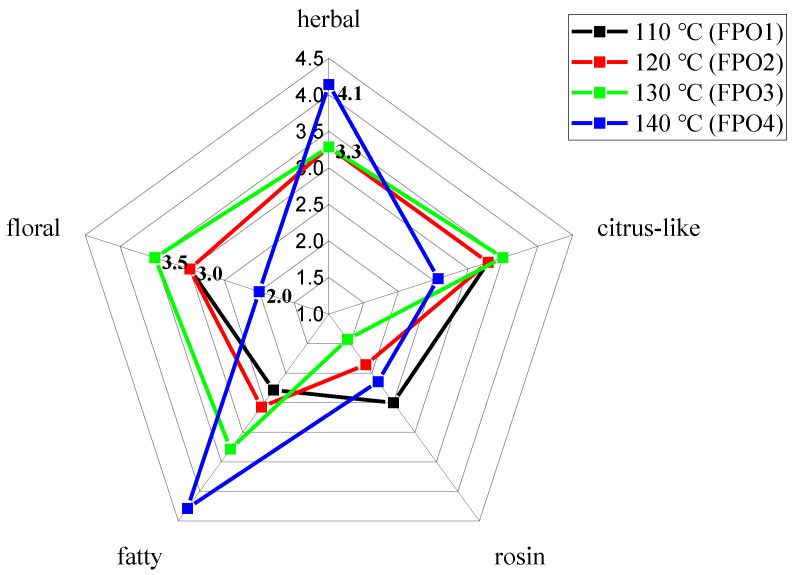
Aroma profiles of FPOs with different frying temperatures.

**Figure 2 foods-11-01661-f002:**
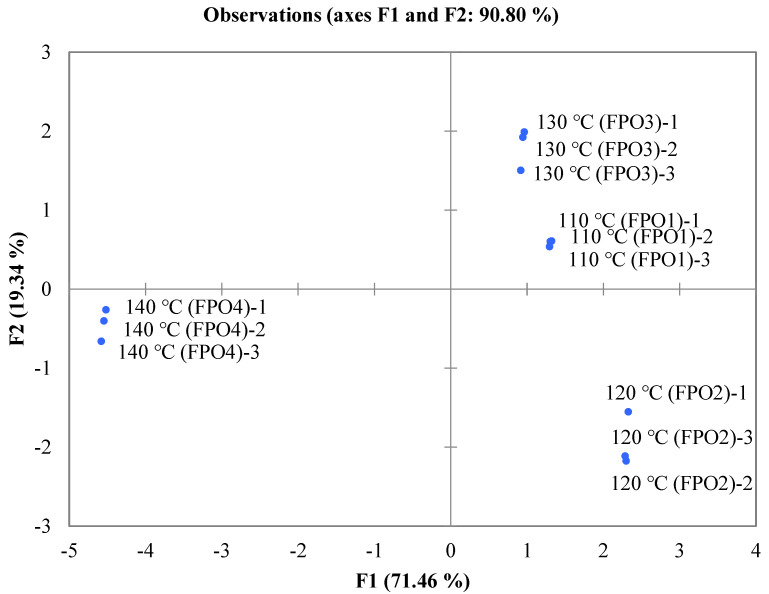
PCA chart for aroma profiles of FPOs with different frying temperatures.

**Figure 3 foods-11-01661-f003:**
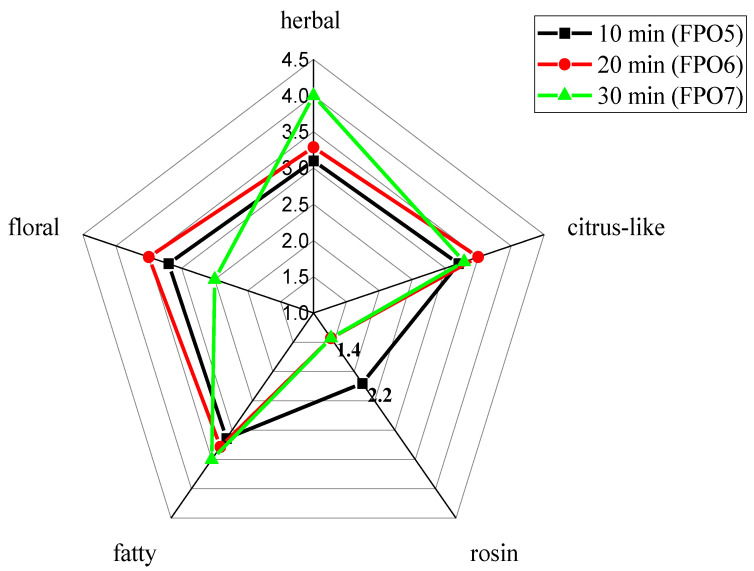
Aroma profiles of FPOs with different frying times.

**Figure 4 foods-11-01661-f004:**
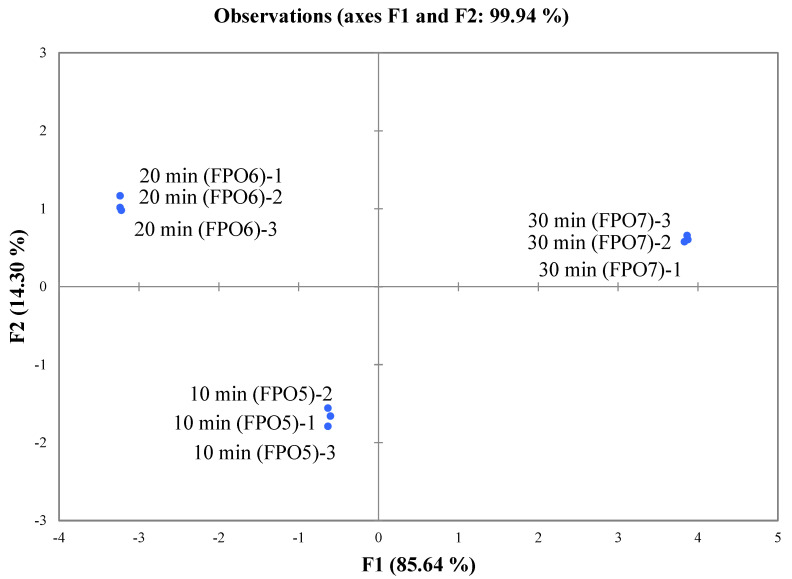
PCA chart for aroma profiles of FPOs with different frying times.

**Figure 5 foods-11-01661-f005:**
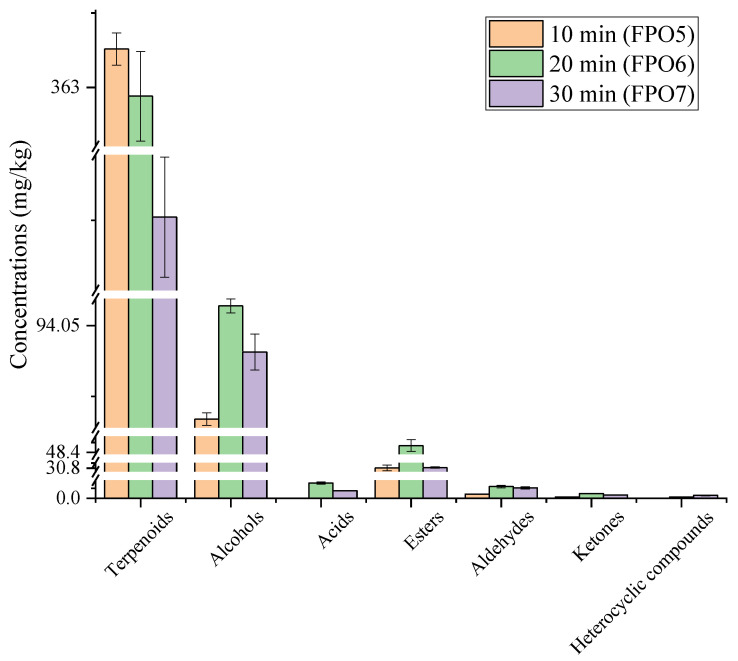
Volatile compound concentrations of FPOs with different frying times.

**Figure 6 foods-11-01661-f006:**
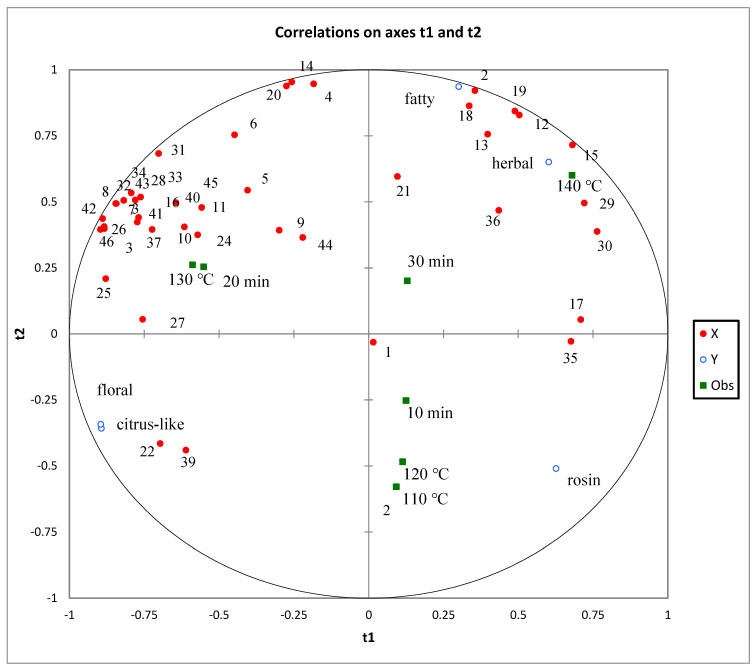
PLSR correlation loading plot of the relationships between 46 volatile compounds (red plots) and sensory attributes.

**Table 1 foods-11-01661-t001:** Volatile compounds of pepper oil prepared at different frying temperatures by GC–MS.

NO.	Compound	Concentrations (mg/kg)				Identification
		110 °C (FPO1)	120 °C (FPO2)	130 °C (FPO3)	140 °C (FPO4)	
1	Sabinene	0.57 ± 0.08 ^a^	0.3 ± 0.02 ^b^	0.49 ± 0.05 ^a^	0.5 ± 0.06 ^a^	MS,RI,Std
2	*β*-Myrcene	28.48 ± 1.2 ^a^	26.84 ± 1.49 ^a^	83.45 ± 3.85 ^b^	38.53 ± 6.58 ^c^	MS,RI,Std
3	Copaene	ND ^a^	ND ^a^	0.18 ± 0.01 ^b^	ND ^a^	MS,RI,Std
4	Hydroxyacetone	ND ^a^	ND ^a^	0.16 ± 0.02 ^b^	0.19 ± 0.03 ^b^	MS,RI,Std
5	Limonene	121.1 ± 2.63 ^a^	183.99 ± 3.85 ^b^	261.13 ± 5.45 ^c^	218.24 ± 5.48 ^d^	MS,RI,Std
6	1,8-Cineole	3.14 ± 0.59 ^a^	2.27 ± 0.12 ^b^	5.95 ± 0.11 ^c^	4.53 ± 0.84 ^d^	MS,RI,Std
7	(*E*,*E*)-2,4-Hexadienal	ND ^a^	ND ^a^	0.44 ± 0.04 ^b^	ND ^a^	MS,RI,Std
8	(*E*,*Z*)-2,6-Dimethyl-2,4,6-octatriene	0.03 ± 0 ^a^	ND ^b^	2.53 ± 0.12 ^c^	0.04 ± 0 ^a^	MS,RI
9	*β*-Ocimene	1.39 ± 0.28 ^a^	3.46 ± 0.16 ^b^	3.29 ± 0.05 ^b^	3.03 ± 0.51 ^c^	MS,RI,Std
10	*δ*-3-Carene	0.11 ± 0.02 ^a^	0.02 ± 0 ^b^	7.15 ± 0.14 ^c^	0.1 ± 0.02 ^a^	MS,RI
11	*p*-Cymene	0.34 ± 0.06 ^a^	0.28 ± 0.03 ^b^	1.44 ± 0.06 ^c^	0.47 ± 0.11 ^c^	MS,RI,Std
12	Terpinolene	ND ^a^	ND ^a^	0.04 ± 0 ^b^	0.13 ± 0.02 ^c^	MS,RI,Std
13	Octanal	ND ^a^	ND ^a^	0.08 ± 0.01 ^b^	0.18 ± 0.02 ^c^	MS,RI,Std
14	(*E*)-2-Heptenal	ND ^a^	ND ^a^	0.88 ± 0.05 ^b^	0.92 ± 0.03 ^b^	MS,RI,Std
15	*β*-Terpineol	ND ^a^	ND ^a^	0.34 ± 0.01 ^b^	ND ^a^	MS,RI,Std
16	Perillen	0.55 ± 0.09 ^a^	0.28 ± 0.01 ^b^	0.21 ± 0.03 ^c^	0.59 ± 0.11 ^a^	MS,RI,Std
17	2-Acetylfuran	ND ^a^	ND ^a^	0.08 ± 0.01 ^b^	031 ± 0.01 ^c^	MS,RI,Std
18	(*E*)-Limonene oxide	0.41 ± 0.06 ^a^	0.36 ± 0.01 ^b^	0.49 ± 0.14 ^a^	0.77 ± 0.05 ^c^	MS,RI,Std
19	1-Octen-3-ol	ND ^a^	ND ^a^	0.14 ± 0.01 ^b^	0.14 ± 0.02 ^b^	MS,RI,Std
20	5-Methyl furfural	ND ^a^	0.13 ± 0.01 ^b^	0.13 ± 0.01 ^b^	0.23 ± 0.05 ^c^	MS,RI,Std
21	Linalool oxide	1.79 ± 0.33 ^a^	1.2 ± 0.03 ^b^	1.83 ± 0.07 ^a^	ND ^c^	MS,RI,Std
22	(*E*,*E*)-2,4-Heptadienal	0.13 ± 0.02 ^a^	0.2 ± 0.01 ^b^	1.04 ± 0.15 ^c^	2.54 ± 0.42 ^d^	MS,RI,Std
23	Perillalcohol	0.13 ± 0.02 ^a^	ND ^b^	ND ^b^	ND ^b^	MS,RI,Std
24	Dihydrocarvone	0.07 ± 0.01 ^a^	ND ^b^	0.16 ± 0.02 ^c^	ND ^a^	MS,RI,Std
25	*β*-Caryophyllene	ND ^a^	ND ^a^	0.54 ± 0 ^b^	ND ^a^	MS,RI,Std
26	Linalool	44.56 ± 2.61 ^a^	65.29 ± 1.09 ^b^	85.12 ± 1.01 ^c^	30.98 ± 0.78 ^d^	MS,RI,Std
27	Linalyl acetate	9.18 ± 1.56 ^a^	7.17 ± 0.77 ^b^	47.33 ± 1.07 ^c^	15.33 ± 1.88 ^d^	MS,RI,Std
28	(*E*)-2-Decenal	ND ^a^	ND ^a^	ND ^a^	0.23 ± 0.05 ^b^	MS,RI,Std
29	Hotrienol	0.48 ± 0.11 ^a^	0.41 ± 0.01 ^a^	0.28 ± 0.01 ^b^	1.16 ± 0.21 ^c^	MS,RI,Std
30	(*E*)-p-2,8-Menadien-1-ol	0.14 ± 0.02 ^a^	0.1 ± 0.01 ^b^	0.34 ± 0 ^c^	0.21 ± 0.03 ^d^	MS,RI
31	4-(1-Methylethyl)-2-cyclohexen-1-one	ND ^a^	ND ^a^	0.44 ± 0.02 ^b^	ND ^a^	MS,RI
32	Terpinyl acetate	0.16 ± 0.02 ^a^	ND ^b^	1.56 ± 0.14 ^c^	0.46 ± 0.06 ^d^	MS,RI,Std
33	*α*-Terpineol	ND ^a^	ND ^a^	1.38 ± 0.04 ^b^	0.2 ± 0.04 ^c^	MS,RI,Std
34	(+)-Isomenthol	0.73 ± 0.12 ^a^	0.34 ± 0.02 ^b^	0.25 ± 0.01 ^c^	0.73 ± 0.13 ^a^	MS,RI
35	(2*E*,4*E*)-2,4-Decanedienal	ND ^a^	ND ^a^	ND ^a^	0.55 ± 0.12 ^b^	MS,RI,Std
36	Neryl acetate	ND ^a^	ND ^a^	0.41 ± 0.03 ^b^	ND ^a^	MS,RI,Std
37	Geranyl acetate	0.17 ± 0.04 ^a^	ND ^b^	0.6 ± 0.06 ^c^	0.2 ± 0.03 ^a^	MS,RI,Std
38	Piperitone	0.28 ± 0.06 ^a^	0.17 ± 0.02 ^b^	0.26 ± 0 ^a^	ND ^c^	MS,RI,Std
39	Carveol	0.59 ± 0.12 ^a^	0.26 ± 0.01 ^b^	0.99 ± 0.02 ^c^	0.46 ± 0.09 ^d^	MS,RI,Std
40	(-)-Carveol	ND ^a^	ND ^a^	0.84 ± 0.04 ^b^	ND ^a^	MS,RI,Std
41	Phenylethyl alcohol	ND ^a^	ND ^a^	0.77 ± 0.1 ^b^	ND ^a^	MS,RI,Std
42	*p*-1,8-Menadien-7-ol	ND ^a^	ND ^a^	0.39 ± 0.02 ^b^	ND ^a^	MS,RI
43	2-Acetyl pyrrole	ND ^a^	ND ^a^	0.19 ± 0.01 ^b^	ND ^a^	MS,RI,Std
44	Octanoic acid	ND ^a^	ND ^a^	1.4 ± 0.12 ^b^	ND ^a^	MS,RI,Std
45	Nonanoic acid	ND ^a^	ND ^a^	1.48 ± 0.14 ^b^	ND ^a^	MS,RI,Std
46	Decanoic acid	ND ^a^	ND ^a^	0.47 ± 0.01 ^b^	ND ^a^	MS,RI,Std

“ND”: volatile compounds not detected; “Std”: confirmed by authentic standards; “MS”: Identification based on NIST 14 mass spectral database; “RI”: RIs on TG-Wax. Means within different letters are significantly (*p* < 0.05) different for the same parameter. Tukey’s post hoc test (*p* < 0.05) was performed to compare means and samples that were significantly different.

**Table 2 foods-11-01661-t002:** Volatile compounds in pepper oils with different frying times as determined by GC–MS.

NO.	Compound	Concentrations (mg/kg)			Identification
		10 min (FPO5)	20 min (FPO6)	30 min (FPO7)	
1	Sabinene	0.67 ± 0.05 ^a^	0.49 ± 0.05 ^b^	0.41 ± 0.05 ^b^	MS,RI,Std
2	*β*-Myrcene	91.22 ± 1.94 ^a^	83.45 ± 3.85 ^b^	71.52 ± 2.71 ^c^	MS,RI,Std
3	Copaene	ND ^a^	0.18 ± 0.01 ^b^	ND ^a^	MS,RI,Std
4	Hydroxyacetone	ND ^a^	0.16 ± 0.02 ^b^	0.09 ± 0.01 ^c^	MS,RI,Std
5	Limonene	270.88 ± 1.39 ^a^	261.13 ± 5.45 ^a^	198.79 ± 10.99 ^b^	MS,RI,Std
6	1,8-Cineole	4.2 ± 0.39 ^a^	5.95 ± 0.11 ^b^	6.27 ± 0.44 ^c^	MS,RI,Std
7	(*E*,*E*)-2,4-Hexadienal	0.08 ± 0.01 ^a^	0.44 ± 0.04 ^b^	0.04 ± 0 ^c^	MS,RI,Std
8	(*E*,*Z*)-2,6-Dimethyl-2,4,6-octatriene	0.09 ± 0.01 ^a^	2.53 ± 0.12 ^b^	0.08 ± 0.01 ^a^	MS,RI
9	*β*-Ocimene	0.97 ± 0.06 ^a^	3.29 ± 0.05 ^b^	1.59 ± 0.18 ^c^	MS,RI,Std
10	*δ*-3-Carene	5.16 ± 0.02 ^a^	7.15 ± 0.14 ^b^	6.24 ± 0.03 ^a^	MS,RI,Std
11	*p*-Cymene	1.41 ± 0.03 ^a^	1.44 ± 0.06 ^a^	1.32 ± 0.04 ^b^	MS,RI,Std
12	Terpinolene	0.05 ± 0.01 ^a^	0.04 ± 0.01 ^a^	0.06 ± 0.01 ^a^	MS,RI,Std
13	Octanal	0.1 ± 0.01 ^a^	0.08 ± 0.01 ^a^	0.09 ± 0.01 ^a^	MS,RI,Std
14	(*E*)-2-Heptenal	0.24 ± 0.01 ^a^	0.88 ± 0.05 ^b^	0.51 ± 0.07 ^c^	MS,RI,Std
15	*β*-Terpineol	ND ^a^	0.13 ± 0.01 ^b^	0.1 ± 0.01 ^b^	MS,RI,Std
16	Perillen	0.58 ± 0.05 ^a^	0.34 ± 0.01 ^b^	0.65 ± 0.08 ^c^	MS,RI,Std
17	2-Acetylfuran	ND ^a^	0.08 ± 0.01 ^b^	ND ^a^	MS,RI,Std
18	(*E*)-Limonene oxide	0.49 ± 0.01 ^a^	0.49 ± 0.14 ^a^	0.53 ± 0.12 ^a^	MS,RI,Std
19	1-Octen-3-ol	ND ^a^	0.14 ± 0.01 ^b^	0.1 ± 0.01 ^b^	MS,RI,Std
20	5-Methyl furfural	ND ^a^	0.13 ± 0.01 ^b^	ND ^a^	MS,RI,Std
21	Linalool oxide	1.91 ± 0.15 ^a^	1.83 ± 0.07 ^a^	1.91 ± 0.29 ^a^	MS,RI,Std
22	(*E*,*E*)-2,4-Heptadienal	0.41 ± 0.03 ^a^	1.04 ± 0.15 ^b^	1.17 ± 0.14 ^b^	MS,RI,Std
23	Dihydrocarvone	ND ^a^	0.16 ± 0.06 ^b^	ND ^a^	MS,RI,Std
24	*β*-Caryophyllene	ND ^a^	0.54 ± 0 ^b^	ND ^a^	MS,RI,Std
25	Linalool	64.64 ± 0.65 ^a^	85.12 ± 1.01 ^b^	75.36 ± 2.82 ^c^	MS,RI,Std
26	Linalyl acetate	30.68 ± 0.64 ^a^	47.33 ± 1.07 ^b^	30.65 ± 0.14 ^a^	MS,RI,Std
27	(*E*)-2-Decenal	0.09 ± 0.01 ^a^	ND ^b^	ND ^b^	MS,RI,Std
28	Hotrienol	0.78 ± 0.06 ^a^	0.28 ± 0.01 ^b^	1.24 ± 0.16 ^c^	MS,RI
29	(*E*)-p-2,8-Menadien-1-ol	0.17 ± 0.01 ^a^	0.34 ± 0 ^b^	0.2 ± 0.01 ^a^	MS,RI
30	4-(1-Methylethyl)-2-cyclohexen-1-one	ND ^a^	0.44 ± 0.02 ^b^	0.35 ± 0.03 ^b^	MS,RI
31	Terpinyl acetate	ND ^a^	1.56 ± 0.14 ^b^	ND ^a^	MS,RI,Std
32	*α*-Terpineol	0.22 ± 0.01 ^a^	1.38 ± 0.04 ^b^	0.18 ± 0.01 ^a^	MS,RI,Std
33	(+)-Isomenthol	0.82 ± 0.05 ^a^	0.25 ± 0.01 ^b^	1.02 ± 0.11 ^c^	MS,RI
34	(2*E*,4*E*)-2,4-Decanedienal	ND ^a^	ND ^a^	0.47 ± 0.03 ^b^	MS,RI,Std
35	Neryl acetate	ND ^a^	0.41 ± 0.03 ^b^	0.07 ± 0.01 ^c^	MS,RI,Std
36	Geranyl acetate	0.16 ± 0 ^a^	0.6 ± 0.06 ^b^	0.19 ± 0.01 ^a^	MS,RI,Std
37	Piperitone	0.3 ± 0.02 ^a^	0.26 ± 0 ^a^	0.3 ± 0.03 ^a^	MS,RI,Std
38	Carveol	0.64 ± 0.06 ^a^	0.99 ± 0.02 ^b^	0.79 ± 0.08 ^c^	MS,RI,Std
39	(-)-Carveol	ND ^a^	0.84 ± 0.04 ^b^	0.46 ± 0.02 ^c^	MS,RI,Std
40	Phenylethyl alcohol	ND ^a^	0.77 ± 0.1 ^b^	ND ^a^	MS,RI,Std
41	*p*-1,8-Menadien-7-ol	ND ^a^	0.39 ± 0.02 ^b^	0.58 ± 0.03 ^c^	MS,RI
42	2-Acetyl pyrrole	ND ^a^	0.19 ± 0.01 ^b^	0.63 ± 0.08 ^c^	MS,RI,Std
43	Octanoic acid	ND ^a^	1.4 ± 0.12 ^b^	1.7 ± 0.04 ^c^	MS,RI,Std
44	Nonanoic acid	ND ^a^	1.48 ± 0.14 ^b^	ND ^a^	MS,RI,Std
45	Decanoic acid	ND ^a^	0.47 ± 0.01 ^b^	ND ^a^	MS,RI,Std

“ND”: volatile compounds not detected; “Std”: confirmed by authentic standards; “MS”: Identification based on NIST 14 mass spectral database; “RI”: RIs on TG-Wax. Means within different letters are significantly (*p* < 0.05) different for the same parameter. Tukey’s post hoc test (*p* < 0.05) was performed to compare means and samples that were significantly different.

## Data Availability

The data showed in this study are contained within the article.

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
