# Peer review of "Effects of Frying Conditions on Volatile Composition and Odor Characteristics of Fried Pepper (Zanthoxylum bungeanum Maxim.) Oil"

_foods, 2022, doi:10.3390/foods11111661_

Round 1

Reviewer 1 Report

The publication is well written and clear. The authors raise interesting topics of practical importance.

Line 13 and 14: "FPO3" and "FPO6" these abbreviations are used in the introduction but not explained.

This article does not provide information on:

a) identification strategies and criteria for RI and MS (match factor), Std

b) characteristics of the quantitative method (LOD, calibration curve, etc.)

Some of this data can be included in supplementary materials

What is "Std" in Tables 1 and 2? What is the ± error in the table - SD, RSD ...?

Some qualitative results were previously published by the authors in [4], but there is hardly any reference to this work in the Results section.

Figure 5 - maybe a better choice would be to log the concentration in this figure?

Reviewer 2 Report

Fried paper oil (FPO) is interesting for Chinese cuisine, and the objective of this study was to evaluate effects of the frying process on the aroma and sensory profiles of FPO by DSA, E-nose and SAFE-GC-MS analysis.

It is in my opinion that the manuscript foods-1743558 entitled "Effects of Frying Conditions on Volatile Composition and Odor Characteristics of Fried Pepper (Zanthoxylum bungeanum) Oil" needs some revisions.

The manuscript has some positive components, but the manner in which the manuscript is written does not meet high standards of the journal.

Technical quality is questionable since it is not clear how many frying repetition is made for each treatment. It seems that mean is calculated on the basis of three repetition of the same sample analysis.

Sensory analysis is made in incorrect way for sensory analysis of oil, provided in PET bottles and with only 1g of samples that is really low amount for sensory analysis. Training session needs to be explained, since trained panel members are mentioned. In addition, in sensory analysis description should be added that only odour was scored.  STDs in sensory analysis are not visible in Figures 1 and 3.

In addition, each legend of the Figure and Tables need to be improved to be self-explanatory.

In the discussion are discussed FPO1-FPO7 treatments, and in the figures and tables are only temperature and time are visible, and therefore, it is very difficult to follow discussion (names of treatments should be same in the tables/figures and discussion). When sensory analysis is connected to results of volatile composition (discussion of results), reference to figure need to be done.

Title: add Maxim in Pepper (Zanthoxylum bungeanum Maxim.)

In the abstract, meaning of FPO1-7 should be added.

Please use E- or trans- in whole manuscript but uniform, not both.

In Abstract and Page 6, line 219:  (E,E)-2,4-decdienal – please check the right name of volatile. It should be same as mentioned in the tables.

Page 3, Lines 93-95:

-          add: …were used in this investigation.

-          Explain purpose of corn oil.

-          Chose one name of oil and use in the manuscript: corn oil or corn germ oil.

Please use aroma and odour instead of flavour, since flavour means odour and taste.

Page 5, line 193: please explain more clear which results are consistent with results of sensory analysis. Page 6, line 205: please use more probability during the discussion of results (one of the reasons for strongest citrus-like aroma…, probably, etc.), since issue of volatile is very complex.

Line 209, delete would add increased,…decreased.

Reviewer 3 Report

Thank you for submitting the manuscript “Effects of Frying Conditions on Volatile Composition and Odor Characteristics of Fried Pepper (Zanthoxylum bungeanum) Oil” to Foods. Overall the article is well designed and the analyzes are complementary with respect to the information. In my opinion, some corrections need to be made.

Line#10: Briefly describe the processing conditions that varied.

Line#23: avoid words that are in the title

Lines#27-28: from what? incomplete sentence.

Lines#26-33: The sentences are too short and although they are on the same subject they are disconnected. Rephrase.

Line#42: please start a new paragraph after the period.

Overall, the entire introduction needs to be reworked as the sentences are short and often incomplete from a language point of view.

Figure 1: Correct lines.

Line#164: I can't define where there is a significant difference or not in this figure. Please add some marker to the values where there is a significant difference.

Figure 2: Say what the abbreviations are. I don't understand what scent it is. PCA is normally used to relate more than one dimension, an example would be to relate the results shown in figure 1 with the results obtained by e-nose. As it stands, Fig 2 confirms what Figure 1 had already shown: that the profiles are different. But the relationship of the two analyzes would indicate how much and why they are different.

Line#206: simple?

Line#224: please cite figure 1 or say in which analysis the fatty aroma appears

Merge figure 2 and 4.

Discussion is too short. Parts of the results item are for discussion. Line#300 for example is about discussion. Please correct all text.

Round 2

Reviewer 2 Report

Even improvement of the manuscript “Effects of Frying Conditions on Volatile Composition and Odor Characteristics of Fried Pepper (Zanthoxylum bungeanum Maxim.) Oil” was made after revision, still some improvement to reach high standard of Foods journal should be done.

Statistical analysis (ANOVA) on the data of volatile compounds and sensory analysis should be done that treatments could be compared and differences among treatments statistically confirmed. If some other conclusion after statistical analysis are reached, discussion should be revised accordingly.

Regarding Tables and Figures – there are still not descriptive enough to be self-explanatory, statistical approach, number of repetition, ect. For example, Figure 5 title: “Concentrations with different frying times.” – it is not known what kind of compounds, how many repetition, statistical approach is missing, SD should be visible in the Figure 5. Other tables and figures titles and legends also should be revised.

Regarding sensory analysis of oil, for the next experiment, you can consider way of sensory analysis of virgin olive oils. Glasses made of glass are more appropriate for sensory analysis, and higher volume should be used (for example 15 mL).

Paragraph 2.7.-2.9. Regarding analysis of volatile compounds could be combined in one paragraph.

Abstract: It is still not visible, which time and temperature is used in experiment, please add.

Line 23. Use “ould be reason” instead of “was reason”.

Regarding the oil sensory analysis, for the future experiment, more sample volume should be used and glasses made of glass to avoid other odours, please consider standard method for virgin olive oil for example.  
